# Chemical, Physical, and Hydraulic Properties as Affected by One Year of *Miscanthus* Biochar Interaction with Sandy and Loamy Tropical Soils

**Sara de Jesus Duarte [1],\*** , **Bruno Glaser [2]** , **Renato Paiva de Lima [3]** and **Carlos Eduardo Pelegrino Cerri [1]**

1   Soil and Plant Nutrition, Luiz de Queiroz College of Agriculture, University of São Paulo, Avenida Pádua Dias, 11-Agronomia, Piracicaba, SP 13418-900, Brazil; cepcerri@usp.br

2   Soil Biogeochemistry, Martin Luther University Halle-Wittenberg, Von-Seckendorff-Platz 3, 06120 Halle, Germany; bruno.glaser@landw.uni-halle.de

3   Agricultural Engineering, Federal Rural University of Pernambuco, Rua Dom Manoel de Medeiros, s/n, Dois Irmãos, CEP, Recife, PE 52171-900, Brazil; renato_agro_@hotmail.com

\*   Correspondence: saraduarte@usp.br

**Abstract:** Biochar application has improved soil properties contributing to crop growth. This study evaluates the effect of biochar amount on soil physical, chemical and hydraulic properties in sandy (SD) and clay loam (CL) soils under tropical conditions. An incubation experiment was installed under laboratory conditions with eight treatments (control, two kinds of soils, SD and CL, and three biochar doses (6.25, 12.5, and 25 Mg ha$^{-1}$). Analyses of soil water retention, bulk density (BD), total porosity (TP), pores size, total carbon (TC), and N were performed after one year. The BD slightly decreased by 0.035 and 0.062 Mg m$^{-3}$ and TP increased by 1.87 and 2.31% in CL and SD soil respectively, upon 6.25 to 25 Mg ha$^{-1}$ biochar application. TC increased in CL and SD by 6.5 and 4.2 kg kg$^{-1}$, respectively, compared to control. The total nitrogen content increased upon biochar addition in CL soil than in SD soil. We found a positive effect of biochar on water availability, microporosity, and a small effect on water retention, especially for CL soil at high biochar application, but this influence did not occur for SD, possibly due to the short time of interaction.

**Keywords:** biochar amount; soil physics; soil fertility; soil water

## 1. Introduction

Traditionally, many approaches have been used to increase the agricultural production, such as the application of fertilizers, organic material, and irrigation. Recently, based on the black earth in the Brazilian Amazon region [1], many researchers have studied biochar and its ecological functions in order to improve soil fertility, soil physical properties, and mitigation of greenhouse gas emissions [2]. Due to these functions, biochar is an attractive tool in agriculture for carbon sequestration and soil quality improvement [3].

Diversified functions in one product are possible because the biochar structure is composed of condensed aromatic groups that give the biochar its black color. Aromatic C groups in the biochar structure provide stability in soils, which makes biochar a good option for long-term C storage in soils [4]. Biological degradation of biochar leads to partial oxidation of biochar functional groups, resulting in the formation of functional groups on the edges of biochar, causing reactivity in the soil such as nutrient retention and mineral-organic stabilization [4,5]. In addition large porosity and specific surface area, and small density of biochar contribute to increased soil water storage and porosity

and decreased bulk density of soils, which are particularly important in compacted and degraded soils [6,7].

When biochar is present in a soil system, it can enhance water storage by modifying the portion of the soil pore size distribution associated with aggregation improvements [8]. Water retention is possible because biochar is a porous material and has the potential to absorb and retain large amounts of water [9]. The capability of biochar to retain water is a function of the combination of its porosity and surface functionality. Glaser et al [2] demonstrated that 18% higher water retention capacity in biochar-amended soils relative to adjacent soils containing low amounts of biochar. Additionally, other soil physical properties can be improved, such as bulk density, porosity, soil aggregation, and soil compaction [10]. These effects may enhance the water availability to crops and decrease soil erosion.

Biochar may not only change soil physical properties but also affect soil chemistry by improving soil fertility and long-term C storage, thus leading to multiple benefits regarding climate change mitigation and adaptation [2]. Due to these beneficial effects, biochar can be used as a soil amendment for improving the quality of agricultural soils [2].

The application of biochar to soils is considered as a win-win strategy to increase soil C sequestration [11]. Biochar is a key component of a potentially sustainable integrated agronomic-biomass-bioenergy production system [11]. Improving soil fertility and soil physical conditions that influence soil hydraulic parameters as water retention [6,7,12,13].

Although there is a wealth of studies concerning the effect of biochar on ecosystem functions [2,14–16], relatively little is known on the effect of biochar on soil physical and hydraulic conditions and the influence of biochar amount. The hypotheses of this paper is that with increase of biochar amount the physical, chemical and hydraulic properties will be improved. The purpose of this paper is to evaluate the effect of biochar amount on the soil physical, chemical, and hydraulic properties of sandy and clay loam soil under tropical conditions.

## 2. Material and Methods

### 2.1. Biochar Preparation

Biochar was commercially produced from agricultural residues of *Miscanthus giganteus*. This biochar was used because it is a co-product of the production of cellulosic ethanol and has been widely applied because of its increasing availability as a product of the growing biofuels industry. The production of biochar involved drying of the *Miscanthus* grass at 120–130 °C in a reactor, followed by pyrolyzation in a second reactor at 450 °C for about 15 min.

### 2.2. Experimental Setup

The factorial experimental design was completely randomized and comprised the factors biochar rate (0, 6.25, 12.5, and 25 Mg ha$^{-1}$) and soil texture (sand and clay loam), totaling eight different treatments, with four replicates, resulting in a total of 32 pots that were incubated for 12 months at temperatures of 30 °C under laboratory conditions. Each container was composed of 100 g of soil and the respective dose of biochar.

### 2.3. Sample Preparation, Incubation, and Sampling

Biochar was incorporated into the soil, based on pre-calculated bulk densities, which included the contribution of the amendments to the final bulk density. The incorporation occurred into a container of 0.5 L at a rate of 0, 6.25, 12.5, and 25 Mg ha$^{-1}$. The interaction soil-biochar occurred in one year, after this, the soil was sampled. For the sampling, a ring of approximately 7 cm$^3$ was used. The collection of the sample occurred with the insertion of the ring in the soil and the removal of the sample with the advice of aluminum similar to one shovel.

*2.4. Biochar and Soil Analysis*

2.4.1. Biochar Analysis

The elemental analyses as well as pH and electrical conductivity and yield were performed following the methods recommended by the International Biochar Initiative Guideline [17], labile and stable carbon [18], moisture content [19], volatile matter [20] and ash content [21], and specific surface area [22].

2.4.2. Fourier Transform-Infrared Spectroscopy (FTIR)

This analysis was conducted in order to verify absorption bands associated with hydrophobic and hydrophilic groups of biochar. FTIR was used as a suitable tool to visualize and quantify aromatic, carboxylic and phenolic groups at the biochar surface. The biochar was ground until the particle size less than 0.149 mm, the biochar and KBr were dried 24 h at 105 °C and 700 °C, respectively. Sonogel pastilles were prepared at 0.1% of biochar and 99.9% of KBr. In this study, a Nicolet Nexus 670 FTIR (Thermo Electron Corporation, Verona Road Madison, WI, USA) was used. All samples were analyzed in transmittance. FTIR imaging mode while collecting all FTIR spectra in the mid-infrared range from 400 to 4000 cm$^{-1}$, using 4 cm$^{-1}$ spectral resolution, and 100 scans for each single point collection.

2.4.3. Hydrophobicity

Considering that water wets soil material with a 90° contact angle, we used an index of water repellency (molarity method of ethanol drop MED) proposed by [23]. This procedure employs the concept that a liquid can only completely enter the soil if the theta (θ) is less than 90°. The procedure evolved drying the biochar in air and putting approximately 2 g of the sample on the ring (50 mm × 10 mm) with the surface of the biochar planed. Solutions of H$_2$O:EtOH with 95% of ethanol were prepared with concentrations of EtOH: 100, 50, 30, 20, 10, and 0. One drop (0.05 mL) of these solutions was applied on the surface of the biochar at a height greater than 5 mm. The time of complete drop penetration in the sample was measured, the temperature at the moment of the test was 25 °C, and the test was repeated with incremented the solution molarity in 1% until that the drop enters in one time smaller than three seconds. Using one graphic which correlates the percentage of the volume of ethanol that penetrate in the biochar less than three seconds with the surface of tension. We found the surface of tension in each sample and the molarity of the ethanol for each sample [24]. We follow the classification of severity of water repellency where, soils with a MED index ≤ 1 not water repellent, 2 very low, 3–5 low, 6–8 moderate, 9–10 severe, and 10–12 very severe [24].

2.4.4. Soil Analyses

The soils used were Arenosol (sandy texture) and Ferralsol (loamy texture). The soils were sampled at the 0–20 cm layer from two different native vegetation areas located near Anhembi, Brazil (22°43′31.1″ S and 48°1′20.2″ W) and in Piracicaba, Brazil (22°42′5.1″ S and 47°37′45.2″ W), both in Sao Paulo state, Brazil. The samples were air-dried, homogenized, and sieved <2 mm.

Soil chemistry characterization as pH in calcium chloride 0.01 M (CaCl$_2$), P, K, Ca, and Mg in resin, H + Al potential acidity pH SMP; Al extracted with potassium chloride (KCl, 1M), sulfate (S-SO$_4$) extracted with calcium phosphate Ca(H$_2$PO$_4$)$_2$, 0,01M), percentage of saturation by bases (V%) and percent saturation by aluminum (m%) [25], C and N was determined with elemental analyzer [26].

*2.5. Physical Analysis*

2.5.1. Bulk Density and Particle Density

The bulk density was determined by the method of [27]. For the determination of particle density (PD) we used approximately one gram of each soil treated with biochar and the control (only soil).

The soil was placed inside of the instrument and the particle density was determinate using a helium gas pycnometer, model ACCUPYC 1330 (Micrometrics Instrument Corporation, Nacross, GA, USA). The pycnometer determines the volume of solids, by the variation of the pressure of one gas, in a known volume chamber.

### 2.5.2. Soil Texture

Soil texture was analyzed in four replicates, since the maximum dose (25 Mg ha$^{-1}$) is the one that would be most likely to change the texture of the soil, we tested only that dose and the control treatment (only soil) by the method of [28] that use sodium hexametaphosphate for dispersion. Sand and silt were obtained by wet sieving and clay by sedimentation.

### 2.5.3. Porosity

The total porosity (TP) was calculated from the soil bulk density (BD) and the particle density (PD) [29] using the following equation:

$$TP = \frac{PD - BD}{BD} * 100 \tag{1}$$

The macroporosity, mesoporosity, and microporosity were calculated by soil water retention curve using theoretical values for macroporosity superior to 50 μm, mesoporosity between 15 and 50 μm, and microporosity less than to 15 μm [30].

### 2.5.4. Water Retention Curve

Soil water holding capacity was measured by moisture contents of the samples at different matric potentials ($-15$, $-10$, $-3$, $-1$, $-0.3$, $-0.1$, $-0.06$, $-0.04$, and $-0.02$ bar). The points $-15$, $-10$, $-3$, $-1$, $-0.3$, and $-0.1$ bar in Richard chamber, and $-0.06$, $-0.04$, and $-0.02$ bar in Haines' apparatus. Gravimetric analysis was undertaken to determine the moisture contents and these were converted to volumetric basis using the corresponding bulk density values [31] and for the curve was used Van Genuchten type [32].

### *2.6. Data Analyses*

Statistical analyses and the graphics were performed using RStudio [33] and the biochar spectrum in the software Origin Lab version 9.1. The data were checked for normality and homogeneity of variances to meet the assumptions of ANOVA and Tukey's test, and a probability of 0.05.

## 3. Results

### *3.1. Soil and Biochar Characterization*

According to Agronomic Institute of Campinas (IAC) and Brazilian Agricultural Research Corporation (Embrapa) the sandy soil has very low amount of K, low of Mg, P, and base saturation (V%), medium of Ca, S, Al, CEC, and aluminum saturation (m%) and very acidic pH. The loamy soil has a neutral pH, low Al, and m%, medium S and CEC, high V% and K, and very high C (Table 1).

Chemical and physical properties of biochar and specific surface area, electric conductivity, pH and total nitrogen and carbon are given in Table 1. Biochar has a high quantity of C, N, Ca, Mg, Na, K, P, and S compared to the soil (Table 1).

The FTIR spectrum of the *Miscanthus* biochar can be recognized in the first main region and the second region. The first region occurs in the interval between 0 and 2000 cm$^{-1}$ contains individual signals and the second signal in 3685 cm$^{-1}$ (Figure 1).

**Table 1.** Physical and chemical characteristics of biochar *Miscanthus*, sandy and loamy soil *.

| Property | Biochar | Sandy Soil | Loamy Soil |
|---|---|---|---|
| | | (%) | |
| Sand | | 90 | 40.6 |
| Silt | | 2.2 | 27.7 |
| Clay | | 7.8 | 31.7 |
| pH ($H_2O$) | 5.86 | | |
| pH ($CaCl_2$) | | 3.90 | 6.50 |
| Yield 31 | | | |
| Moisture | 3.5 | | |
| Volatile material | 34.6 | | |
| Ash | 6.1 | | |
| Fixed carbon | 57.1 | | |
| Total N | 0.43 | | |
| $^{13}C$ | −13.41 | | |
| Total C | 66.35 | 0.86 | 1.93 |
| N | 0.43 | 0.06 | 0.17 |
| C/N | 155.4 | 14.3 | 11.4 |
| Labile C (%) | 2.70 | | |
| Stable C (%) | 50.87 | | |
| Lability | 0.05 | | |
| CEC ($mmol_c dm^{-3}$) | 33 | 69 | 138 |
| Specific surface area ($m^2\ g^{-1}$) | 371.9 | | |
| Electric conductivity (($\mu S\ cm^{-1}$) | 605 | | |
| | | (mg kg$^{-1}$) | |
| Na | 423 | | |
| P | 1859 | 4 | 28 |
| S | 634 | 5.30 | 9.50 |
| Fe | 1317 | | |
| Mn | 139 | | |
| Cu | 59 | | |
| Mo | 0.66 | | |
| Zn | 138 | | |
| Ni | 7.74 | | |
| | | ($mmol_c dm^{-3}$) | |
| Al | | 5.7 | 0 |
| H + Al | | 62 | 18 |
| SB | | 6.9 | 120 |
| V (%) | | 10 | 87 |
| m (%) | | 45 | 0 |

* Source: [34]. CEC: cation exchange capacity. H + Al = potential acidity, SB = soma of bases (Ca, Mg, K), CEC = Cation exchange capacity (SB + Al + H), V = Base saturation (SB × 100/CTC); m = Aluminum saturation (100 × $Al^{3+}$/ SB + $Al^{3+}$).

The first signal in the points 500 can be associated to carbonate (C-O), 860, 894 cm$^{-1}$ can be attributed to cyclic acid anhydrides (C-C; C-O) group; the second one at 644 and 1095 associated with the band 1030 cm$^{-1}$ is due to silicon (Si-O), the signal at 1448, 1484, and 1606 can be assigned to carboxylates asymmetric ($CO_2$) and at 1700 and 1780 can be attributed to ketones (C=O) and aromatic carbon, respectively. The second region is due to the presence of silanol (Si-O-H) at 3685 cm$^{-1}$.

Among these compounds, are hydrophobic: Cyclic acid anhydrides (C-C and C-O) carboxylates asymmetric ($CO_2$), ketones aromatic (C=O), Silicon (Si-O) and hydrophilic carbonates (C-O) and silanol (Si-O-H). Although the hydrophilic compounds (silanol and carbonates) have high intensity, the quantity of hydrophobic compounds with high intensity is larger in this biochar (Figure 1). These compounds may contribute to the high hydrophobicity in this biochar, this hydrophobicity

was verified by analyze of the biochar hydrophobicity in which the result was high hydrophobicity (Figure 2).

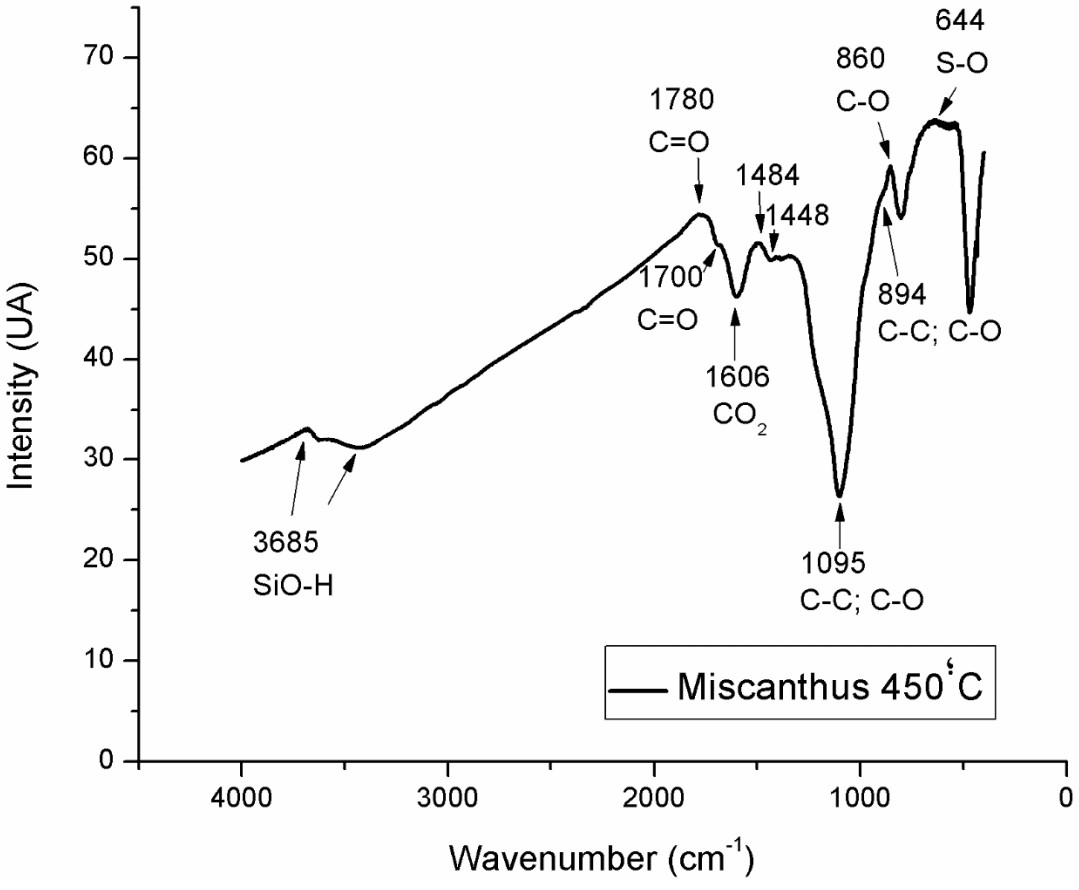

**Figure 1.** Fourier transform infrared spectrum of *Miscanthus* biochar pyrolyzed at 450 °C.

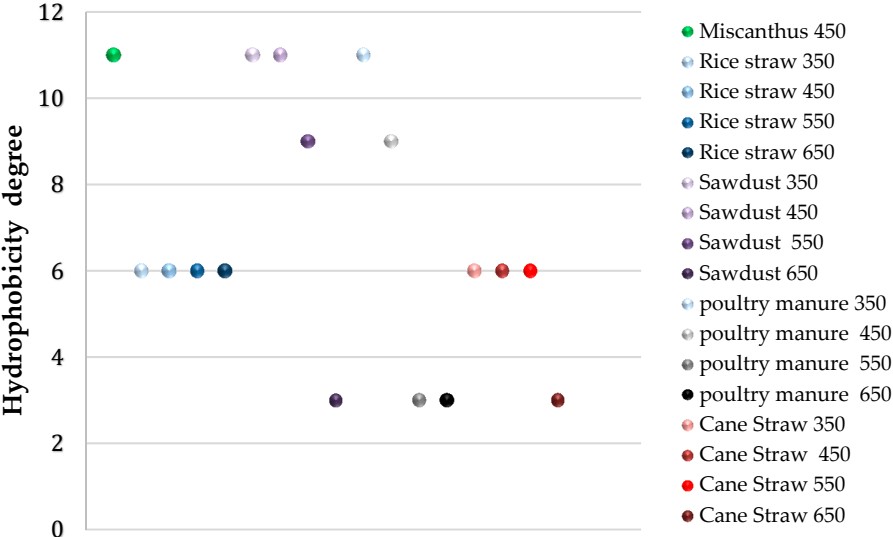

**Figure 2.** Hydrophobicity *Miscanthus* biochar pyrolyzed at 450 °C the scale 0–12 is indicative of degree of hydrophobicity (≤1 not water repellent; 2: Very low; 3–5: Low; 6–8: moderate, 9–10: Severe; 11–12: very severe).

## 3.2. Effect on Soil Chemical Properties

### 3.2.1. Total Carbon

As expected we found higher amounts of carbon in the loamy soil compared to the sandy soil (Table 1). The total carbon increased with biochar addition in both soils, but for loamy soil, this effect was more accentuated than for sandy soil. In the loamy soil, the total organic C ranged from 23.1 g kg$^{-1}$ at the control to 29.6 g kg$^{-1}$ at the highest biochar addition (25 Mg ha$^{-1}$). This dose increased the amount of carbon by 6.6 g kg$^{-1}$ in loamy soil (Figure 3A). For the sandy soil, the total organic C ranged from 5.7 g kg$^{-1}$ at the control treatment to 10 g kg$^{-1}$ at 25 Mg ha$^{-1}$ of biochar addition from the soil. Therefore, this increase was of 4.2 g kg$^{-1}$.

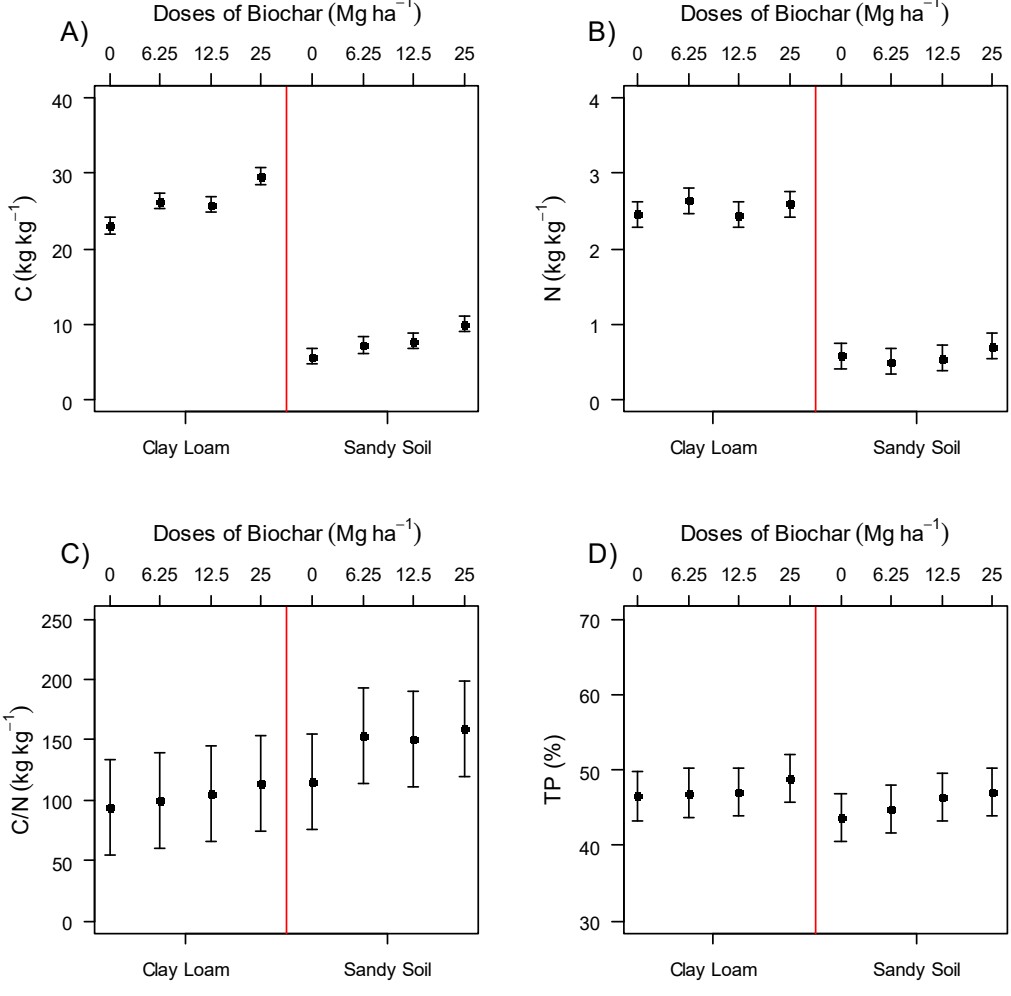

**Figure 3.** *Cont.*

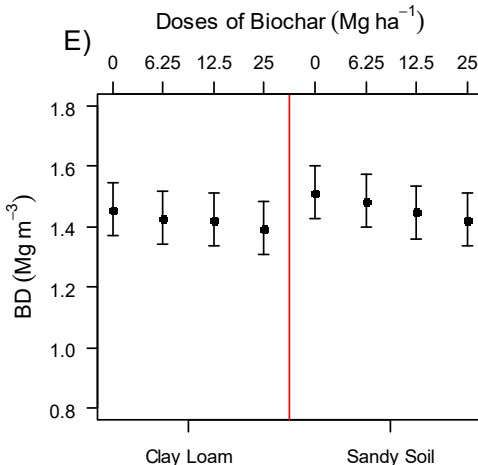

**Figure 3.** Effect of biochar amount on soil physical and chemical properties in clay loam and sandy soil. C: carbon; N: nitrogen; C/N: atomic ratio; TP: total porosity; BD: bulk density. (**A**) Carbon content; (**B**) Nitrogen content; (**C**) Carbon and Nitrogen ratio; (**D**) Total porosity; (**E**) Bulk density.

### 3.2.2. Total Nitrogen

Total N (TN) content ranged between 2.4 g kg$^{-1}$ and 2.6 g kg$^{-1}$ for the loamy soil and between 0.58 and 0.70 g kg$^{-1}$ for sandy soil. (Figure 3B). The increasing amount of biochar application tended to increase soil TN contents compared to the control, but this difference was not statistically significant ($p > 0.05$).

### 3.2.3. C:N Ratio

For the loamy soil the C/N ratio ranged between 9.4 (control) and 11.4 (25 Mg ha$^{-1}$ of biochar; Figure 3C); for the sandy soil, C/N ratio ranged between 11.6 (control) and 15.9 (25 Mg ha$^{-1}$ of biochar; Figure 3C). Increasing biochar amounts increased the C/N ratio, but this increase was not statistically significant ($p < 0.05$).

### 3.3. *Effect on Soil Physical Properties*

### 3.3.1. Porosity

The biochar amount increased the porosity in both soils. Although the difference is not significant is possible to note that in both soils the biochar amount increased the total porosity. Compared to the control treatment in the loamy soil, the 25 Mg biochar ha$^{-1}$ increased the porosity from 47 to 49% and for sandy soil from 44 to 47% (Figure 3D).

### 3.3.2. Bulk Density

There is no significant difference ($p > 0.05$) between loamy and sandy soil and biochar doses on bulk density. In both soils, the total bulk density decreased with increasing of biochar, but for sandy soil, this effect was more accentuated than for loamy soil. In the loamy soil, the bulk density ranged from 1.46 g cm$^{-3}$ at the control to 1.39 g cm$^{-3}$ in 25 Mg ha$^{-1}$ (Figure 3E) and for sandy soil between 1.51 g cm$^{-3}$ (control) and 1.42 g cm$^{-3}$ (25 Mg ha$^{-1}$).

### 3.3.3. Texture

The biochar addition did not change the soil texture in both sandy and clay loam soil, even when applied 25 Mg ha$^{-1}$ of biochar (Table 2).

**Table 2.** Influence of biochar amount on soil texture in sandy and clay loam soil.

| Treatment | Sand | Silt | Clay | Textural Class |
|---|---|---|---|---|
| | | (%) | | |
| Control | 40.6 | 27.7 | 31.7 | Clay Loam |
| Biochar 25 Mg ha$^{-1}$ | 34.7 | 27.0 | 38.3 | Clay Loam |
| Control | 90 | 2.2 | 7.8 | Sandy |
| Biochar 25 Mg ha$^{-1}$ | 91 | 5.8 | 3.3 | Sandy |

Control = only soil.

### 3.4. Effect on Soil Hydraulic Properties

3.4.1. Water Retention Curve

In the loamy soil at low biochar doses (6.25 and 12.5 Mg ha$^{-1}$), low tension was enough to reduce the water content of the macropores. However, in control treatment, this decrease occurred from 100 hPa and, for the highest biochar amount (25 Mg ha$^{-1}$), was necessarily approximately 1000 hPa, for the decrease the water content. We checked that the higher biochar amount increased the water retention and was higher than the other treatments up to 1000 hPa and from that potential water retention was similar for all treatments (Figure 4a).

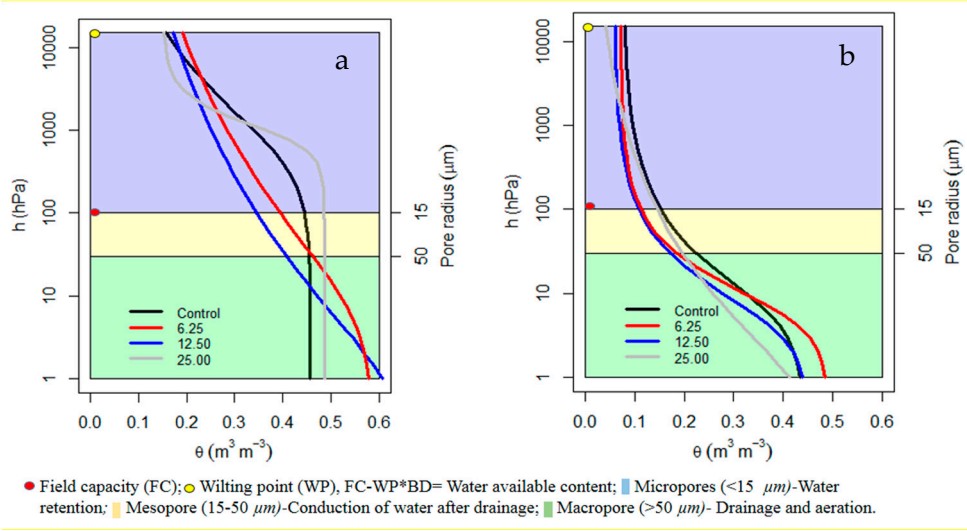

**Figure 4.** Effect of biochar amount on soil water retention in clay loam and sandy soil. (**a**) Clay loam soil; (**b**) Sandy soil.

The biochar amount did not affect the water retention in sandy soil, little increase in the soil water retention in low potential (1–10 hPa) was checked in 12.5 Mg ha$^{-1}$. From of potential (10 hPa), the biochar dose did not affect the soil water retention, the curve behavior was similar in all treatments and the difference between the treatments in the water retention was very little and insignificant (Figure 4b).

3.4.2. Plant-Available Water Content

Comparing loamy and sandy soil, the clay loam soil has higher water available content than the sandy soil. In clay Loam soil and sandy soil the higher biochar dose increased the water available content compared to control treatment in 0.05 cm$^3$cm$^3$ in clay loam and 0.03 cm$^3$ of water per cm$^3$ of soil in sandy soil (Figure 5).

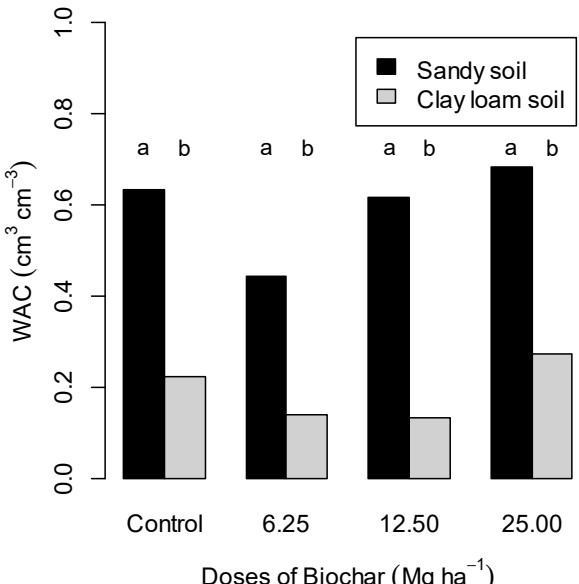

**Figure 5.** Effect of biochar doses on plant-available water content in clay loam and sandy soil. AWC Plant available water content.

### 3.4.3. Effect of Biochar Doses on Pore Size Distribution and Its Relation with the Water Retention Curve

Biochar application not only increased the total porosity but also alters soil pore size distribution and water retention. The biochar doses affected the macroporosity, mesoporosity, and microporosity in both soils, the changes in the porosity altered the behavior in the water retention curve for both soil and in all treatments (Figure 6).

In the loamy soil, in the treatments control and 25 Mg ha$^{-1}$, the total porosity was approximately 48 and 49% respectively. The very low quantity of macropores kept the same behavior in the curve. The variation occurred only in the microporosity range from 100 hPa, the high water retention in these treatments could be attributed to microporosity in the control and 25 Mg ha$^{-1}$ of biochar. Different behavior in the water retention curve occurred for 6.25 and 12.5 Mg ha$^{-1}$ of biochar; the macroporosity was higher compared to other treatments. In the 6.25 Mg ha$^{-1}$ dose, 61% of the pores comprised approximately 38% of macropores, 9% mesopores, and 53% micropores, and the high quantity of macropores was responsible for the high drainage of water at a low potential. In the 12.5 Mg ha$^{-1}$ dose, 59% of the total porosity comprised 21% of macropores, 12% mesopores, and 67% micropores.

The intermediate doses of biochar (6.25 and 12.5 Mg ha$^{-1}$) increased the macroporosity and mesoporosity and decreased the microporosity. This porosity affected the curve in low potential. From 100 hPa, the field capacity increases the water retention with an increase of the tension in all treatments of the clay loam soil (Figure 6).

In sandy soil, the macroporosity was high in all treatments. This macroporosity contributed to high drainage at low matric potential, up to 100 hPa the low quantity of micropores promoted little changed in the water retention curve. In control treatment, 44% of the pores was controlled by macroporosity, 17% by mesopores, and 35% by micropores. In the 6.25 and 12.5 Mg ha$^{-1}$ doses, the volume of approximately 24% of micropores contributed to the low variation in the water content when increased the potential. The addition of 25 Mg ha$^{-1}$ of biochar increased the microporosity to approximately 30% (Figure 6).

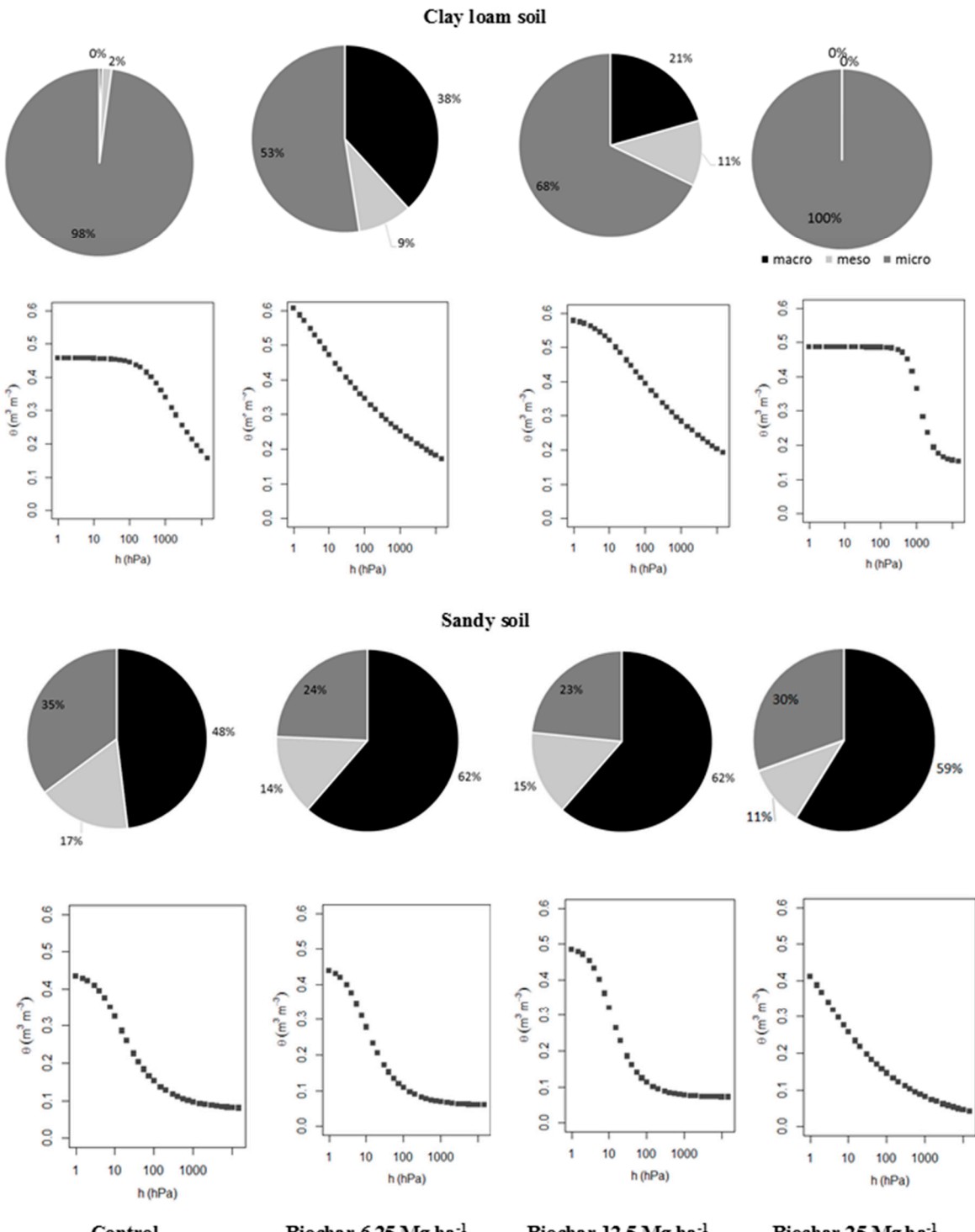

**Figure 6.** Effect of biochar amount on macropores (>50 μm); mesopores (15–50 μm) and micropores (<15 μm), and its relation with water retention.

## 4. Discussion

During temporal biochar exposure to the environment, hydrophobic moieties such as polycondensed aromatic moieties decrease while hydrophilic functional groups such as carboxylic acids increase [2,5].

Both soils have low fertility but the added biochar has high fertility (Table 1). Therefore, biochar has a direct positive effect of the investigated tropical soils. It was already stated by [2] that a potential fertilization effect of biochar is high. Laird at al [14] observed significant increases in plant-available P, K, Ca, and Mn upon 5 and 10 Mg ha$^{-1}$ biochar addition to the soil after 500 days. This increase was attributed to the presence of these nutrients in the biochar itself that over the time can be available in the soil. Because of chemical and biological biochar stability, the biochar is less reactive in soil than other organic molecules the biochar degradation requires a long time. The study of [35] revealed a biochar degradation of 25% within 100 years, translated into a biochar turnover of around 300 years.

*4.1. Biochar Effects on Soil Nutrient Storage Capacity*

4.1.1. Total Carbon

The addition of 25 Mg ha$^{-1}$ of biochar increased the total carbon in both soils but this difference was significant only for clay loam soil compared to lower doses and the control. A similar result in clay loam soil was verified by [15] after one year of interaction (Figure 3A). Similarity, [14] verified that the addition of biochar to the loamy soil increased the total C content after the 500 days incubation by 17.6, 37.6, and 68.8%, respectively, for the 5, 10, and 20 g kg$^{-1}$ biochar treatments in relation to control treatment (only soil).

Sandy soil has a tendency to increase carbon content with biochar addition (6.25 to 25 Mg ha$^{-1}$), but this difference was not significant (Figure 3A). Liu et al [6] reported a three-fold increase of soil organic carbon upon 20 Mg ha$^{-1}$ biochar addition to sandy soil. Weber et al [36] reported higher C and N contents three years after compost with biochar application from municipal waste in the range of 30 to 120 Mg ha$^{-1}$ at a field trial with Dystric Cambisol.

The increase of carbon concentration of 6.6 kg kg$^{-1}$ in the loamy soil and 4.2 kg kg$^{-1}$ in sandy soil with addition of 25 Mg ha$^{-1}$ (Figure 3A) can contribute for increase of aggregation stability, water retention, water available content, and reduction of soil bulk density, these soil physical properties are essential for the soil physical quality and, therefore, for plant development.

4.1.2. Nitrogen

Increase in the quantity of nitrogen with an increase of the biochar doses was also verified by [2,15] in clay loam soil. As well as we found in this experiment, that occurred increase but the difference was not significant with one year of interaction, [12] verified that the difference was not significant in the first year of soil-biochar interaction.

The time required for the increase of the nitrogen in the soil with biochar addition can be attributed to the N in the biochar is present in a stable form [14]. Therefore, the increase of 0.2 and 0.12 g Kg$^{-1}$ for loamy and sandy soil respectively (Figure 3B), can increase with the time of interaction. The increase of nitrogen amount in the soil is essential for the plant productivity, the biochar addition can increase the amount of nitrogen in the soil without the need of the farmer spend a lot of money with fertilizers.

4.1.3. C:N Ratio

Independent of biochar amount, the loamy soil has lower C:N ratio compared to sandy soil, which can be attributed to the high quantity of nitrogen in clay loam soil, the increase in the biochar dose increased the C:N ratio, for sandy soil, increased 4.3% and for clay loam soil 2%. This increase can be attributed to the high quantity of carbon in biochar (Figure 3C). As well, in this article, Zhang et al [15] verified that the high amount of biochar amendment of 40 Mg ha$^{-1}$ induced a high C:N ratio 14–16 under 40 Mg ha$^{-1}$ as compared to 13 under 20 Mg ha$^{-1}$. The low increase in the C:N ratio can be associated with low dose in both soils. The increase in the C:N ratio decreased the decomposition of the material if the C:N ratio is superior to 35, the decomposition is low, in every treatment the C:N ratio was inferior to 21, indicated that we do not have problems with decomposition and nitrogen immobilization [37].

*4.2. Biochar Effects on Soil Physical Properties*

4.2.1. Total Porosity

Similar to our results, many studies have proved that biochar addition increased the porosity in clay loam and sandy soil [7]. The increase of the porosity with an increase of biochar amount occurred because the biochar is a material with high total porosity, composed by approximately 80% of pore volume, with normally, 95% of these pores are less than a 20 nm of diameter [38], when the biochar is putted in the soil the total porosity increase (Figure 3D). The high porosity in biochar can be a strategy for increasing the porosity, aeration and water movement, water holding capacity, heat and gases in degraded soils, and with compaction problems [12]. The alteration in the porosity of the soil is reflected in the bulk density, but it is important to consider that the low bulk density and high total porosity may not be always beneficial for plant growth, because of slight compaction can improve root-soil contact and pore connectivity, allowing higher nutrient and water transport and supply to plant roots [39].

4.2.2. Bulk Density

Biochar addition to soil reduced the bulk density even at low amount (Figure 3e), in a few cases, biochar application at rates <10 Mg ha$^{-1}$ [40] may not reduce bulk density. However, in most research even in small quantities biochar reduces bulk density, and with increase the biochar amount the decrease in the bulk density is more expressive [12,41–43]. Omondi et al [7] reviewed studies on biochar and soil bulk density prior to 2016, and found that the biochar application reduced bulk density by 3–31% in 19 out of 22 studies, indicating that bulk density generally decreases with biochar application.

The magnitude of the biochar effect on bulk density can be explained by simple dilution of the soil with the low bulk density of the biochar [14]. This is so important for the reduction of bulk density and increases soil porosity that contributes directly to the plant development. Similar to this result (Figure 3e), Blanco-Canqui [12] found that the effect of biochar on a reduction of bulk density on clayey soils can be smaller than sandy soil, because of larger differences in size and density between biochar and sand particles being larger than those between biochar and clay particles. The biochar could also reduce bulk density in the long term by interacting with soil particles and improving aggregation and porosity favoring the water retention and availability of water for the plants and promoting a suitable means for the plant development [12]. These results proved that in clay loam and sandy soil the farmers can use biochar for improve the soil physical properties such as bulk density and porosity, properties necessary for the increase the soil quality and consequently plant productivity.

4.2.3. Texture

Our results suggest that in clay loam and sandy soil, biochar addition up to 25 Mg ha$^{-1}$ did not modify soil texture. Similarly, Githinji [43] found the same result up to 25 Mg ha$^{-1}$, but with 50 Mg ha$^{-1}$ he verified that the soil texture shifted slightly from 'loamy sand' to 'sand'. This can occur due to the presence of many larger particles in the range of sand in the biochar.

*4.3. Biochar Effects on Soil Hydraulic Properties*

4.3.1. Water Holding Capacity

The fact that the biochar has been produced from Si-enriched raw feedstock (Figure 1), can increase the water retention in the soil because the Si reacts with water molecules in polymerizations reactions causing phytolith or silica hydrogel formations, which are essential structures used in plant biochemical and biophysical reactions [44]. The result is that the biochar may express the same tendency to react with soil water by physically adhering water molecules or trapping water vapor in internal pores [45]. Soon, assuming that water-binding pathways of Si-enriched biochar operate in

soils, the Si content of biochar may be an important characteristic to improve soil-water storage, and therefore, the feedstock switchgrass can be used for improving the soil water retention [12]. Though the hydrophobicity in this biochar can contribute to the reduction of the efficiency in the improvement of soil water retention, the contact with water by means of the saturation process for the production of the retention curve and the maintenance of the field capacity of the samples during one year of soil-biochar interaction can reduce this hydrophobicity. Kinney [46] affirmed that the exposition of the biochar in water can reduce its hydrophobicity.

Considering that the biochar is hydrophobic when the contact angle is higher than 90° [47] observed a 69.5% decrease of the contact angle of biochar after one year of its addition to soil suggesting that initial biochar hydrophobicity disappeared within one year. This decrease in hydrophobicity will improve soil water retention [46]. Based on these results it is possible that the hydrophobicity of this biochar used in this experiment was not a negative factor for water retention.

The effect of biochar on water retention is dependent on soil texture [48]. In the loamy soil, the water retention at low doses (6.25 and 12.5 Mg ha$^{-1}$) of biochar was smaller than the treatment with no addition of biochar (Figure 4). Two reasons can explain this behavior. The first one is that smaller biochar doses inferior to 20 Mg ha$^{-1}$ have no effect in the water retention [49] or can reduce the water retention because of fine-textured soils may be less responsible to biochar application and larger amounts of biochar may be needed to increase water retention which can result in increased plant available water [12]. Medium (20 Mg ha$^{-1}$) and high (100 Mg ha$^{-1}$) biochar applications can improve water-holding capacity as verified by [50]. The second reason is that in those doses (6.25 and 12.5 Mg ha$^{-1}$) the organic material was higher and differed from the control treatment. Peake et al [50] affirmed that soils rich in organic material responded less or not to biochar amendment. Devereux Devereux et al [42] showed that an increase in biochar amount results in an increase in soil water content for a given matric potential, suggesting that as matric potential increases, biochar retains more water within pores as compared to biochar-free sandy loam soil.

The high water retention at 25 Mg ha$^{-1}$ of biochar can be explained by the high dose applied also the high specific surface area in biochar (Figure 4) that associated to the high the electric conductivity can contribute for increasing the water retention in biochar and consequently increase the soil water storage. Blanco-Canqui [12] found in the literature review that the biochar increases the ability of the soil to retain water in 90% of cases but the large amounts of biochar can be required to increase water retention consistently. Finally, in the control treatment, the high microporosity can be associated with high soil bulk density (Figure 3e), which can contribute for the increase in water retention in this treatment.

In sandy soil, with pour organic material, we not verified improvement in the soil water retention, the no effect of biochar on the soil can be attributed to the biochar that was applied alone. Laird et al [14] and Christensen [51] verified that the biochar has no effect in sandy soil if applied alone and suggest the application with organic compost and the incorporation of biochar in the subsoil. Another reason is that the increase of soil water retention can occur in coarse texture soil or soils with a large number of macropores if large amounts of biochar are applied [52]. Gaskin et al [53] reported improvements in soil-moisture storage after biochar was added at high rates (88 Mg ha$^{-1}$) in Ultisol with a coarse texture. In addition, the time of interaction, Salinas et al [54] verified that for increasing the water holding capacity of soils is required a longer period. However, Liu et al [6] reported a two-fold increase of plant-available water holding capacity upon 20 Mg biochar ha$^{-1}$ plus 30 Mg ha$^{-1}$ compost addition to a sandy soil when combined with 30 Mg ha$^{-1}$ compost addition alone.

In the loamy soil, the farmers can use the biochar in the dose 25 Mg ha$^{-1}$ for increase the soil water retention and in sandy soil the same dose is necessary, however, to obtain one good increase in the soil water retention is necessary more than one year of interaction soil-biochar. The increase in the retention of the water is essential for agriculture, ambient and plant productivity. With more water in the soil, the farmers can reduce the frequency of irrigation, reducing the use of water so necessary for the environment. In areas of familiar agriculture that do not use irrigation, the biochar can be used

for provided water in a period drought. With a continuous supply of water to the plants, production tends to grow significantly.

### 4.3.2. Relation between Water and Pore Size Distribution

We found that the biochar application with different particle size changes the soil pore size distribution. This occurred due to intrapores (pores inside biochar particles), most of them having diameters <1 μm (micropores), and the different biochar particle size change the pore space between biochar particles and soil (interpores). The contribution with interpores and intrapores create heterogeneity in pore size distribution, as verified in our results in clay loam and sandy soil. In clay loam soil, intermediate doses (6.25 and 12.5 Mg ha$^{-1}$), increased the macroporosity and mesoporosity and decreased the microporosity, for the control and 25 Mg ha$^{-1}$ the total porosity was dominated by micropores. However, in sandy soil, the macroporosity was high in all treatments, the mesoporosity increased with increase of biochar amount but the microporosity decrease (Figure 6).

The low amount of biochar (6.25 and 12.5 Mg ha$^{-1}$) contributed to increase the macroporosity as verified by Gaskin et al [55]. In sandy soil, similar results were found by [56] who verified that biochar addition increased macropores between 5% and 20%. The increase of macropores (>50 μm) contribute to drainage and aeration of the soil and heat flow [12,48], and the drainage corresponds to decreasing values of water retention and do not hold available water for the plants [56,57] but increases the soil aeration contributing for aerobic microorganisms and plant respiration and the heat flow contributes to warming the soil and the atmosphere near to the soil.

In clay loam soil the mesopores increase only in intermediate doses (6.25 and 12.5 Mg ha$^{-1}$ and decrease in 25 Mg ha$^{-1}$. In sandy soil, only in the high amount 25 Mg ha$^{-1}$ reduced the mesoporosity. The increase in the mesoporosity pores with a diameter between 15–50 μm is important because contribute to conduction of the water in the soil during the water distribution process that occurs after the infiltration, when the macropores are emptied [58].

We found mainly micropores in clay loam soil, at the control treatment and in 25 Mg ha$^{-1}$. The high microporosity, can be associated to high density in the control and high biochar amount in 25 Mg ha$^{-1}$. Due to high microporosity the water retention was high compared to smaller doses (6.25 and 12.5 Mg ha$^{-1}$). Studies have found that biochar can reduce the proportion of drainable pores and increase the quantity of mesopores and micropores (Petersen et al., 2016). The fact that biochar application contribute with pores inside particles (intrapores) less than 1 μm is favorable for the water retention in the soil when matric potential is lower than −165 hPa [58] can explain why only loamy soil with 25 Mg biochar ha$^{-1}$ retained high quantity of water after −165 hPa. However, biochar with a high quantity of pores between biochar particle and soil (intraporosity) is more effective in the water retention at lower soil water potential, these interpores control water retention when soil water potential ($\psi$) is less than −16.5 kPa [58].

The micropores, besides the water retention, increase the plant-available and nutrient storage [59,60]. Cation retention was observed when mixing soil with biochar [2] However, the underlying mechanism for this observation is still unclear. However, the nutrient retention mechanisms, such as pores, surface adsorption, cationic and anionic interaction, are determined by the physical and chemical structure of biochar [2,59,60].

Water availability, storage, and nutrient are crucial for plant productivity. Biochar has the potential to alter soil hydrology and to drive shifts for water stored in soils and increase plant-available water under dry conditions [59,61]. Instead, it would provide more storage of water on the landscape under wet conditions [61]. However, the low quantity of biochar can be favorable for water flow, aeration, soil water retention and water available content for the plants [12,43].

### 4.3.3. Plant-Available Water Content

Similar to this paper, Tryon [61] reported that application of biochar increased plant-available water content in sandy soil and no effect in a loamy soil. Similarly, Blanco- Caqui [12] found that

biochar application consistently increased plant available water in 21 out of the 29 soils, suggesting that plant-available water increased when biochar was applied in 72% of cases such a response may be attributed to the high microporosity of the biochar. Due to the soil moisture, retention may only be improved in coarse-textured soils, a careful choice of biochar/soil combination needs to be taken into consideration.

The increase in plant-available water with biochar suggests that the application of biochar to croplands could contribute to the reduction of the frequency of irrigation. This can be particularly important in water-limited or semiarid regions [2,6].

### 4.3.4. Implications for Farmers in Brazil

It is clear from the present study that high rate of biochar (25 Mg ha$^{-1}$) in clay loam soil can be beneficial for the chemical and physical properties (e.g., C and N concentrations, C/N ratio, plant available water, porosity, and bulk density) improving the soil water content, reduce plant water stress and increase the amount of C and N. However, the good results in the improvement in soil physical and chemical especially in sandy soil, can be associated with time of interaction because the biochar is a stable material [55]. The Brazilian farmers can use the dose 25 Mg ha$^{-1}$ for improvement of physical and chemical properties in sandy and clay loam soil.

### 5. Conclusions

The biochar addition contributes to soil C sequestration. In addition, there was an improvement in the physical properties such as bulk density, and porosity for both for loamy and sandy soil. However, the water retention was different between soils, for the sandy soil the addition of biochar did not improve the water retention, but for loamy soil, biochar increased the water retention when applied at 25 Mg ha$^{-1}$. The maximum dose of biochar altered the pore size distribution, increasing, especially, the mesoporosity and microporosity in clay loam soil and macroporosity in sandy soil.

These results show that biochar can improve soil management in the humid tropics, for example in response to shortages of water. It ought to be possible to recommend to farmers or other land use stakeholders that the biochar with another organic material easily decomposable should be added to a given type of soil with the aim of achieving the desired outcome, e.g., in terms of increase of water retention, the high biochar dose superior to 25 Mg ha$^{-1}$ could be used in clay loam and sandy soil. However, studies are necessary to indicate which is the better dose superior to 25 Mg ha$^{-1}$ for these soils and what is the best organic material that is easily decomposable to include with the biochar.

**Author Contributions:** S.d.J.D. was responsible by experimental conduction, wrote the manuscript and contributed to statistical analyze; B.G. contributed to the review and improvement of the paper. R.P.d.L. contributed to statistical analysis; C.E.P.C. contributed to the experimental design, experimental conduction, and review of the paper.

**Funding:** CNPq (process number 404150/2013-6).

**Acknowledgments:** This study was supported by CNPq (process number 404150/2013-6) and partially developed in University of Sao Paulo Brazil and in the Martin Luther University Halle (Saale), Germany and Fellowship was supported by Coordination for the Improvement of Higher Education Personnel (CAPES).

**Conflicts of Interest:** The authors declare no conflict of interest.

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
