# Peer review of "Chemical, Physical, and Hydraulic Properties as Affected by One Year of Miscanthus Biochar Interaction with Sandy and Loamy Tropical Soils"

_soilsystems, doi:10.3390/soilsystems3020024_

Round 1
Reviewer 1 Report
This is very good paper, well presented and described. In my opinion, it is suitable for publication in its current form after completing with Figures presenting general structures of tables mentioned in the manuscript.
Please make a concise the contents in Table 1 and Table 2.
Please present the title of Figure 5 and x-axis.
Replace Figure 3 and Figure 6 with a high resolution
Author Response
Comments and Suggestions for Authors
This is a very good paper, well presented and described. In my opinion, it is suitable for publication in its current form after completing with Figures presenting general structures of tables mentioned in the manuscript.
Answer: Dear reviewer, thanks very much for your constructive suggestions. We have carefully revised the manuscript. Thanks for that! Please see the revised version of the manuscript and our answers to each one of your comments/suggestions.
Please make a concise the contents in Table 1 and Table 2.
Answer: Thanks, it was done as you suggested.
Please present the title of Figure 5 and x-axis.
Answer: Thanks, it was done as you suggested.
Replace Figure 3 and Figure 6 with a high resolution
Answer: Thanks, it was done as you suggested.

Reviewer 2 Report
In this manuscript entitled “Chemical, physical, and hydrological properties as affected by one year of Miscanthus biochar interaction with sandy and loamy tropical soils”, the authors have evaluated, experimentally, the effect of different doses of biochar addition on physical, chemical and hydric properties of sandy and clay loam soils under tropical conditions.
Analytical data have been collected after an one year interaction of soil and amending Miscanthus biochar. Biochar application had improved soil physical, hydraulic and chemical properties contributing for crop growth.
It is an interesting paper, well written and structured, worth to be published in Soil Systems journal after some minor revisions.
According to the Journal instructions references must be numbered in order of appearance in the text. In the text, reference numbers should be placed in square brackets [ ], and placed before the punctuation.
I think that the term “hydraulic” is more appropriate, referring to soil properties, than “hydric” or “hydrological” throughout the text.
Line 22: You should either define “TN” or replace it by “total nitrogen”.
Table 1: If CEC values presented in the table are original the analytical method should be noted in the Materials and Methods section. Otherwise the relevant reference should be added. Are there any CEC data for the amended soils?
Figure 3 should be improved in terms of fonts or resolution.
Figure 5: Figure caption should be placed below the figure.
Author Response
Comments and Suggestions for Authors
In this manuscript entitled “Chemical, physical, and hydrological properties as affected by one year of Miscanthus biochar interaction with sandy and loamy tropical soils”, the authors have evaluated, experimentally, the effect of different doses of biochar addition on physical, chemical and hydric properties of sandy and clay loam soils under tropical conditions.
Analytical data have been collected after one year interaction of soil and amending Miscanthus biochar. Biochar application had improved soil physical, hydraulic and chemical properties contributing for crop growth.
It is an interesting paper, well written and structured, worth to be published in Soil Systems journal after some minor revisions.
Answer: Answer: Dear reviewer, thanks very much for your constructive suggestions. We have carefully revised the manuscript. Thanks for that! Please see the revised version of the manuscript and our answers to each one of your comments/suggestions.
According to the Journal, instructions references must be numbered in order of appearance in the text. In the text, reference numbers should be placed in square brackets [ ], and placed before the punctuation.
Answer: Thanks, it was done as you suggested.
I think that the term “hydraulic” is more appropriate, referring to soil properties, than “hydric” or “hydrological” throughout the text.
Answer: Thanks, it was done as you suggested.
Line 22: You should either define “TN” or replace it with “total nitrogen”.
Answer: Thanks, it was done as you suggested
Table 1: If CEC values presented in the table are original the analytical method should be noted in the Materials and Methods section. Otherwise, the relevant reference should be added. Are there any CEC data for the amended soils?
Answer: Dear reviewer the CEC values come from another paper Conz et al 2017, I included this information in the paper according to your suggestion. Unfortunately, we don’t have CEC data for the amended soils.
Figure 3 Should be improved in terms of fonts or resolution.
Answer: Thanks, it was done as you suggested
Figure 5: Figure caption should be placed below the figure.
Answer: Thanks, it was done as you suggested
